# Delayed Access to Medical Care and Psychological Distress among Chinese Immigrants in Canada during the Pandemic

**DOI:** 10.3390/healthcare12161639

**Published:** 2024-08-16

**Authors:** Anh Thu Vo, Lixia Yang, Robin Urquhart, Yanqing Yi, Peizhong Peter Wang

**Affiliations:** 1Faculty of Medicine, Memorial University of Newfoundland, St. John’s, NL A1B 3V6, Canada; attvo@mun.ca (A.T.V.); yyi@mun.ca (Y.Y.); 2Department of Psychology, Toronto Metropolitan University, Toronto, ON M5B 2K3, Canada; lixiay@torontomu.ca; 3The Centre for New Immigrant Well-Being (CNIW), Markham, ON L3R 6G2, Canada; 4Department of Community Health and Epidemiology, Dalhousie University, Halifax, NS B3H 1V7, Canada; robin.urquhart@dal.ca; 5Cancer Outcomes Research Program, Nova Scotia Health Authority, Halifax, NS B3S 0H6, Canada; 6Department of Surgery, Nova Scotia Health Authority, Halifax, NS B3H 2Y9, Canada; 7Dalla Lana School of Public Health, University of Toronto, Toronto, ON M5T 3M7, Canada

**Keywords:** rescheduling/cancellation, psychological distress, COVID-19, Chinese immigrants, Canada, LASSO regression

## Abstract

The psychological impact of medical care accessibility during the pandemic has been widely studied, but little attention has been given to Asian immigrants in Canada. This study aimed to fill this literature gap by using a cross-sectional survey, which aimed to evaluate the impact of the COVID-19 pandemic on Chinese immigrants in North America during the second wave of the pandemic. The study focused on Chinese immigrants aged 16 or older in Canada. Covariates included sociodemographic variables, delayed access to medical care (i.e., treatment or health assessment), and other COVID-19 related variables. We used logistic LASSO regression for model selection and multivariate logistic regression models to evaluate the association between delayed access to treatment/health assessment and psychological distress outcome, as measured by the COVID-19 Peritraumatic Distress Index (CPDI). Missing data were handled using multiple imputation. Our study included 746 respondents, with 47.18% in the normal CPDI group and 36.82% in the mild-to-severe CPDI group. Most respondents were originally from Mainland China and residing in Ontario. Over half have stayed in Canada for at least 15 years. The multivariate logistic regression models identified significant risk predictors of psychological distress status: delayed access to medical care (OR = 1.362, 95% CI: 1.078–1.720, *p* = 0.0095), fear of COVID-19 (OR = 1.604, 95% CI: 1.293–1.989, *p* < 0.0001), and social loneliness (OR = 1.408, 95%CI: 1.314–1.508, *p* < 0.0001). Sociodemographic variables and other COVID-19-related variates did not significantly impact the study’s outcome. Our findings shed light on the importance of timely medical care access to psychological well-being among Chinese Canadians. Reliable health information, mental health support, and virtual care tailored to immigrants should be considered to mitigate this impact and promote their overall health and well-being.

## 1. Introduction

During the global COVID-19 pandemic, the Canadian healthcare system faced challenges with insufficient hospital capacity and resources to cope with the increasing demand of hospitalizations and intensive care unit (ICU) admissions due to SARS-CoV-2 [1,2,3]. Many Canadian hospitals faced additional challenges due to shortages of trained healthcare professionals, a lack of upgraded suppliers and systems, and unavailable ICU beds [1,2,3]. 

Given the challenges related to staff/resource shortages, the care of non-COVID-19 patients was significantly interrupted during the pandemic [1,3]. Reallocating staff and resources to care for COVID-19 patients led to many elective surgical procedures and diagnostic services being disproportionately postponed or canceled [1,3]. Additionally, restrictions on healthcare services during the pandemic led to many in-person appointments being delayed or canceled [2,4,5]. Although virtual care endorsement significantly increased during the pandemic [2,6], the services might not be preferred by many patients [2], particularly for “distressed patients”, immunocompromised patients, older people, or those who do not have access to virtual health services [2,5,7].

In addition to various factors such as fear of contracting COVID-19 [8,9] and social determinants of health [10] that resulted in deteriorating psychological status, delayed medical care access also significantly impacted mental health during the pandemic [9,10]. For example, international research showed that the interruption of cancer care delivery resulted in increasing general psychological distress [8,11], posttraumatic stress disorder symptoms [8], and anxiety about negative cancer consequences [12]. Some studies also identified psychological factors considerably associated with changes in access to non-cancer care [13,14,15].

Chinese and other Asian immigrants in North America were more vulnerable to psychological distress because of cultural differences, language barriers, and fears (e.g., fear of stigma, panic attack) during the pandemic [16,17,18,19,20]. Several reports indicated that these populations often perceive or experience discriminatory crimes in various forms, from verbal harassment and racial slurs to physical attacks [17,18,19]. Consequently, many individuals might avoid visiting public places [19,21], perceive or experience unfair treatment [21], and be reluctant to seek care [22], which can make them suffer a higher psychological burden than their counterparts [18,20,21,23].

In Canada, a study revealed that there was a relationship between delayed healthcare services due to some reasons (e.g., fear of COVID-19 virus) and low self-rated mental well-being, as well as perceived declines in mental health [9]. Statistics Canada (2022) showed that vulnerable populations such as those with multiple chronic comorbidities, women, and immigrants were more likely to delay medical care due to fear of contracting the virus [5].

To date, although the impact of changes in access to medical care during the pandemic on psychological factors has been studied by many countries [13,14,15,24], little attention has been given to this specific influence on vulnerable populations such as Asian immigrants in Canada. A dramatic increase in anti-Asian discrimination and socio-economic-related risk factors negatively impacted psychological well-being (i.e., increased distress) among Chinese Canadians during the pandemic [19]. Additionally, disrupted healthcare services may further deteriorate the mental health condition of these populations. This study aims to address the following question: to what extent did delayed medical care access (e.g., treatment, health assessment) impact psychological distress among Chinese immigrants in Canada during the global pandemic? It addresses two specific objectives: (1) to describe and compare the characteristics of Chinese immigrants in Canada by their psychological distress status; (2) to evaluate the association between delayed medical care access and distress levels. Understanding this matter is crucial for informing health authorities and healthcare providers about the negative impacts of compromised or deferred care and treatment on psychological distress among Asian immigrants as a whole and among Chinese immigrants in particular. Moreover, evaluating COVID-19-related disruptions in care delivery through the lens of immigrant populations can provide comprehensive insights into how policy changes and multiple factors impact these vulnerable populations.

## 2. Methodology

### 2.1. Study Design, Setting, and Participants

This study utilized data from a cross-sectional survey collected from 11 January to 28 February 2021. The survey aimed to evaluate COVID-19’s impact on Chinese immigrants and residents in North America during the second wave, including questions on sociodemographic background, perceptions and knowledge of COVID-19, public health practices of COVID-19, multidimensional impacts of COVID-19 on society and healthcare, and behaviors caused by psychological distress.

Through a convenience sampling procedure, Chinese immigrants aged 16 or over, who had lived in Canada for at least 6 months at the time of the survey, were recruited to voluntarily complete this survey. The online survey was developed and delivered through the Qualtrics platform available at Memorial University; it was promoted on various social media platforms, including WeChat, group chat forums, websites of collaborative Chinese community organizations (e.g., CNIW), Chinese media (e.g., OMNI TV), and other approaches (e.g., emails). Responding to the survey was voluntary and anonymous, and participants needed to give online informed consent before beginning the survey. All identifying variables, such as IP address and WeChat ID, were not included in data analysis.

Because we aimed to evaluate the impact of delayed medical care access on the psychological distress of Chinese immigrants in Canada, the final sample included Chinese Canadian citizens and permanent residents aged 16 or older, while excluding other categories of Chinese residents (e.g., international students, family visit/tour, business, or US citizens). Of the 1253 eligible respondents completing the survey, a total of 888 responses met all the inclusion criteria. Among these valid responses, 746 responses with complete psychological distress outcome measures were included in the final data analysis (see Appendix A).

### 2.2. Main Outcome—Psychological Distress

The primary outcome, psychological distress, was assessed with the COVID-19 Peritraumatic Distress Index (CPDI) [25,26]. It includes 24 items assessing distress symptoms, such as “anxiety”, “depression”, “specific phobia”, “compulsive behavior”, as well as “physical and social function symptoms” in the past week. Each item was evaluated using a Likert scale, ranging from “0” for never to “4” for most of the time. CPDI scores were categorized into normal (<28 points), mild to moderate (28–51 points), and severe (≥52 points). This study used the cut-off of mild to moderate to create a binary outcome variable: normal (<28) and mild-to-severe (≥28 points).

### 2.3. Covariates

#### 2.3.1. Sociodemographic Variables

The survey includes the following sociodemographic variables: Age was categorized as under 35, 35 to 44, 45 to 54, 55 to 64, and 65 or above. Gender was reported as men/women or missing (including prefer not answering or non-responses). Birthplace comprised Mainland China and others (including Canada, Hong Kong, Macau, Taiwan, and other regions). Province of residence was categorized as Ontario and outside Ontario. Mother language included Mandarin and others (including Cantonese, English, or other languages). Length in Canada was self-reported: under 6 years, 6 to less than 15 years, at least 15 years. Other sociodemographic backgrounds included marital status (married/common law or others), highest educational status (under university, university, and master/PhD), employment status (employed, self-employed, unemployed, or other), living status (living with family or others), financial status (dissatisfied, neutral, or satisfied), current health status (poor, fair, or good), work in healthcare (yes or no), work requiring public contact (yes or no), and COVID-19 infectious history (yes or no).

#### 2.3.2. COVID-19-Related Variables

Perceived/experienced discrimination was measured from two questions: one assessing the perception of COVID-19-related discrimination (i.e., “Do you think the COVID-19 pandemic has caused discrimination against Chinese immigrants/residents in North America?”) and one assessing the experience of COVID-19-related discrimination (i.e., “Have you experienced discrimination related to COVID-19? If so, what kind of discrimination?”). This binary variable was categorized as “Yes” when the response was “strongly agree” or “agree” and “No” when the response was “neutral”, ”disagree”, or “strongly disagree”.

Two questions were used to evaluate Fear of COVID-19: “Are you personally afraid of contracting COVID-19?” and “Are you afraid that someone in your immediate family will be contracting COVID-19?” Respondents were categorized into two groups: “Yes” for those who were “very” or “somewhat afraid of” COVID-19, versus “No” for those who responded “neutral”, “not very afraid”, or “not afraid at all of” COVID-19.

Loneliness was measured using the De-Jong Gierveld Loneliness Short Scale, a valid and reliable six-item instrument to measure overall, emotional, and social loneliness [27]. The scale has been validated and used in many countries [28], including China [29]. Respondents rated each item on a five-point Likert scale ranging from “1” (strongly disagree) to “5” (strongly agree) [29]. Three items with positive words were reset coded before summing all items to calculate the total score [29]. A higher total score indicates a greater level of loneliness [29].

Delayed medical care access was measured using a question about any experience—whether rescheduling, postponement, or cancellation of surgery, outpatient surgery, follow-up visits for chronic disease, mental illness or common illness, or other health issues or concerns (such as dental, vision, etc.). The question also included cancellations or postponements of other medical consultations with healthcare professionals or routine laboratory tests. Participants who did report any changes in their care were grouped as “Yes”, and all the others were categorized as “No”.

### 2.4. Missing Data

We assumed that missing data in the study were missing at random. All missing data in this study were multiply imputed using “fully conditional specification (FCS)”, which is a valid approach to handling missing data points of categorical and continuous variables [30,31]. Specifically, item-level imputation was used to handle missing data before calculating the CPDI score and social loneliness score. Item-level imputation, which imputes incomplete items prior to calculating a total score, is preferred for its reliable and accurate estimates compared to scale-level imputation [31].

With a missing data rate of 3.38% to 14.30%, we generated 20 datasets for imputation [32,33]. Although incomplete outcome variables were included in the imputation model, we excluded these missing values from the analysis model [32,34]. This approach allowed us to minimize trivial noise in estimates resulting from these values [32,34]. We excluded 15.99% of the imputed outcome values from the final analysis model.

### 2.5. Statistical Analysis

The data analysis was performed using SAS 9.4, and a *p*-value of < 0.05 was considered to be significant for all analysis. For descriptive analysis, the completed dataset was summarized by mean, standard error, confidence interval for continuous variable, and frequency and proportions for categorical variables. The difference of each covariate variable between CPDI groups was examined using Chi-square tests (or Fisher’s exact test when more than 20% of cells with expected cell counts less than five) for categorical variables and the *t*-test for continuous variables.

For logistic regression analysis, univariate logistic regression models were initially used to examine the association between individual covariate variables and CPDI in the completed dataset. For sensitivity analysis, we utilized multivariate regression logistic models for the completed dataset (Completed Case Analysis) and the imputed dataset. We used logistic LASSO regression to select the most significant variables associated with the outcome [35,36]. LASSO penalty function shrinks some coefficients to zero, which may lead to improve prediction accuracy [35,36]. Finally, multivariate logistic regression models were conducted to yield odds ratios in the presence of most significant variables selected from the LASSO model. Moreover, the Hosmer-Lemeshow test was used to check the goodness of fit of the model [37]. The sensitivity and specificity of the model was assessed using the Area Under the ROC Curve, with a cut-point of ROC over 0.7 considered acceptable [37].

## 3. Results

### 3.1. Participants’ Sociodemographic Characteristics

Results from the descriptive analysis present the respondents’ sociodemographic characteristics by the CPDI (see Table 1). Among 746 respondents, there were 419 (47.18%) individuals with normal CPDI and 327 (36.82%) with mild-to-severe CPDI. Participants with mild-to-severe CPDI were more often women (70.59% vs. 62.14%, *p* = 0.0165), individuals from Mainland China (97.55% vs. 93.32%, *p* = 0.0074), respondents with neutral (41.90% vs. 40.19%) or dissatisfied (31.19% vs. 16.99%) financial status (*p* < 0.0001), and those who reported current fair (44.48% vs. 36.84%) to poor (7.06% vs. 2.15%) health status (*p* = 0.0001), compared to those in the normal CPDI group. Most respondents in both groups were married (87.77% in the normal CPDI group vs. 84.97% in the mild-to-severe CPDI group), speaking Mandarin (93.29% vs. 93.8%), residing in Ontario (89.45% vs. 88.62%), and living with their family (91.87% vs. 90.21%). More than half have stayed in Canada for at least 15 years (60.38% in normal CPDI vs. 53.82% in mild-to-severe CPDI). Participants holding a university degree (42.48% in the normal CPDI group vs. 44.04% in the mild-to-severe CPDI group) or master/PhD degree (37.47% vs. 31.19%) are common. There was a high proportion of employed participants in both groups (43.68% normal CPDI group vs. 39.45% mild-to-severe CPDI group), with a small percentage of respondents working in the healthcare sector (9.64% vs. 8.31% in the normal and mild-to-severe groups, respectively) and in workplaces requiring public contact (15.87% vs. 20.25%).

### 3.2. COVID-19-Related Participants’ Characteristics

As outlined in Table 1, a small percentage of participants were infected with COVID-19 (0.48% in normal CPDI vs. 0.61% in mild-to-severe CPDI). A significantly higher prevalence of fear of COVID-19 was observed in those with mild-to-severe CPDI (86.54%) than that in respondents with normal CPDI (60.62%) (*p* < 0.0001). The mild-to-severe CPDI group showed a higher prevalence of delayed medical care access (87.46%) than the normal CPDI group (74.94%) (*p* < 0.0001). Moreover, participants with mild-to-severe CPDI (19.18 ± 2.98) had a higher mean score of social loneliness than their counterparts (16.21 ± 2.73) (*p* < 0.0001). Although the prevalence of perceived or experienced discrimination in the mild-to-severe CPDI group was higher than that in the normal CPDI group (41.90% vs. 35.56%), this difference was insignificant (*p* = 0.0774).

### 3.3. Risk Factors Associated with Mild to Severe COVID-19 Peritraumatic Distress Index (CPDI)

A number of risk factors for mild-to-severe CPDI were identified from a univariate logistic regression model (see Table 2). The odds of mild-to-severe CPDI in men were 31.6% lower than those in women (OR = 0.684, 95% CI: 0.501–0.933, *p* = 0.0167). In contrast, the odds of mild-to-severe CPDI were positively associated with dissatisfied financial status (OR = 2.922, 95% CI: 1.966–4.342, *p* < 0.0001), poor current health status (OR = 4.123, 95% CI: 1.860–9.135, *p* = 0.0026), fear of COVID-19 (OR = 4.178, 95% CI: 2.877–6.068, *p* < 0.0001), delayed medical care access (OR = 2.333, 95% CI: 1.572–3.462, *p* < 0.0001), and social loneliness score (OR = 1.447, 95% CI: 1.354–1.545, *p* < 0.0001).

Next, we utilized multiple imputed datasets to estimate parameter coefficients in multivariable logistic regression analysis. A few risk factors for mild-to-severe CPDI were identified from the multivariate logistic regression model (see Table 3). Selected risk factors were derived from the LASSO logistic regression model, and multivariate logistic regression models were used to yield odds ratios (see Appendix A). In multivariate regression analysis, fear of COVID-19 was positively associated with the odds of mild-to-severe CPDI (OR = 1.604, 95% CI: 1.293–1.989, *p* < 0.0001), indicating that there was a 60.40% increase in the odds of mild-to-severe CPDI if individuals feared contracting COVID-19, compared to their counterparts. The odds of mild-to-severe CPDI in participants who experienced delayed medical care access were 1.362 times higher than those in participants who did not report these experiences (OR = 1.362, 95% CI: 1.078–1.720, *p* = 0.0095).

There was a positive association between the odds of mild-to-severe CPDI and social loneliness score (OR = 1.408, 95%CI: 1.314–1.508, *p* < 0.0001). This means that an increase in social loneliness score of one point was associated with a 40.80% increase in the odds of mild-to-severe psychological distress. However, there was an insignificant association between the odds of mild-to-severe CPDI and sociodemographic covariates (including age, length in Canada, highest educational status, financial status, and current health status).

### 3.4. Sensitivity Analysis

We also conducted Completed Case Analysis (CCA) using completed datasets for a multivariate logistic regression model (Table 3 and Appendix A), which allowed us to identify the influence of missing data on estimates. The results identified fear of COVID-19 (OR = 2.617, 95% CI: 1.694–4.041, *p* < 0.0001), delayed medical care access (OR = 1.900, 95% CI: 1.184–3.048, *p* = 0.0078) and social loneliness score (OR = 1.384, 95% CI: 1.291–1.483, *p* < 0.0001) as positive predictors for mild-to-severe CPDI (see Table 3).

The CCA produced greater odd ratios and 95% confidence intervals for fear of COVID-19 (OR = 2.617, 95% CI: 1.694–4.041) and delayed medical care access (OR = 1.900, 95% CI: 1.184–3.048) compared to the multiple imputation approach (OR = 1.604, 95% CI: 1.293–1.989) and (OR = 1.362, 95% CI: 1.078–1.720), respectively (see Table 3). This discrepancy might be attributed to potential biases and reduced power in the CCA, which resulted from an exclusion of cases with missing data [38]. Sterne et al. (2009) noted that multiple imputation, along with an assumption of missing at random, can correct biases caused by the CCA [38]. As a result, we believe multiple imputation is a valid approach to correct biases caused by missing data points.

## 4. Discussion

The results identified fear of COVID-19, delayed medical care access (e.g., treatment, health assessment), and social loneliness were positively associated with mild-to-severe CPDI among Chinese immigrants in Canada during the pandemic.

### 4.1. Fear of COVID-19 and Psychological Distress

The high prevalence of COVID-19 phobia was reported around the world. Ng et al. (2020) revealed that a high level of COVID-19 phobia was experienced among cancer patients (66%), caregivers (72.8%), and healthcare workers (41.6%) in Singapore [39]. Wang et al. (2020) showed that about 75.2% of the general population of China feared that their family members would be infected by COVID-19 [40]. A 2020 Canadian poll demonstrated a high proportion of fear of contracting COVID-19, with 66% of Canadians concerned about themselves and 76% of Canadians worrying about their family members contracting the virus [41]. We revealed that worries about themselves or their family members contracting COVID-19 are a risk for psychological distress among Chinese individuals in Canada. This finding is aligned with many previous studies’ results [8,42,43,44,45] which indicated a strong relationship between fear of the COVID-19 virus and psychological consequences. Proliferation of negative and unreliable news [44,46], COVID-19-related uncertainties [42,44,46] and having friends or family members infected by the virus or deceased from COVID-19 [43] might exacerbate the fear of COVID-19. Some studies found that adaptive coping approaches, reliable and updated health information, and preventive health measures might mitigate the fear of COVID-19 and psychological distress [40,42,44,47]. Hence, public health agencies play an important role in providing accurate COVID-19-related information to the general population. COVID-19 information from credible sources may enhance understanding of COVID-19 and well-planned preparations to prevent infection, thus ultimately alleviating COVID-19 phobia.

### 4.2. Loneliness and Psychological Distress

Loneliness is a well-known predictor of psychological distress both pre-pandemic [48] and during the pandemic [49,50,51]. Yang et al. (2022) found a negative relationship between loneliness and mental well-being among Chinese immigrants in Canada during the pandemic [20]. Consistent with their findings, the current study has identified the significant impact of social loneliness on psychological distress among this population in Canada during the second wave of the pandemic. Moreover, while unmet mental health needs and seeking mental health support significantly escalated among those experiencing loneliness since the lockdown [52,53], evidence suggested a low rate of mental health services utilization among immigrants due to language barriers, stigma, cultural difference, inadequate health information, and difficulties in navigating the healthcare system [54,55]. It is necessary to promote or develop psychological programs or mental health support tailored to specific immigrant populations.

### 4.3. Delayed Medical Care Access and Psychological Distress

In our study, delayed medical care access (e.g., treatment, health assessment) due to COVID-19 was a predictor of psychological distress status among Chinese immigrants in Canada. Although delayed medical care access resulted in poor mental health among the general population during the pandemic in Canada [9], its impact on specific immigrant populations such as Chinese immigrants has not been examined. Our study filled this literature gap and extended the result to the Chinese immigrant population in Canada. Future studies are needed to explore barriers to care at multiple levels for this population. Virtual care was adopted quickly to mitigate the impacts of interruptions in access to care due to the pandemic. However, Patterson et al. (2022) revealed that infrastructure, regulatory, and coverage policies for these virtual health services were barriers to implementing virtual care [56]. Additionally, older age, challenges in internet use, concerns around personal health privacy and security, and coverage for virtual services were common barriers for Canadian residents when attempting to use virtual care [56]. Brual et al. (2023) also found that virtual care use was lower among older immigrants in Ontario, Canada during the pandemic, particularly those from “family class immigrants” and those with lower competency in English [57]. Policymakers and healthcare providers should consider these factors to improve virtual care services for targeted populations.

### 4.4. Sociodemographic Factors and Psychological Distress

Although birthplace, gender, financial status, and current health status were not significantly associated with mild-to-severe CPDI in multivariate logistic regression analysis, they showed significant effects on psychological distress status in the univariate logistic regression analysis. Additionally, we did not find an association between perceived/experienced discrimination and psychological distress among Chinese immigrants in Canada during the second wave of the pandemic. This finding contrasts with previous studies [20,58], which found a strong influence of discrimination on psychological distress. Hence, these factors should not be negligible when assessing psychological distress among Chinese immigrants in Canada.

Variations in the findings might result from different analytic decisions (e.g., using CPDI as a categorical vs. a continuous variable), the number of covariates and the way to categorize variables, and the chosen study sample [59]. For example, Yang et al. (2022) and Yang et al. (2024) used CPDI as a continuous variable that could result in statistical significance [20,58], thereby reducing type II errors when the continuous variable is treated as a categorical variable [60]. However, our study aimed to examine the risk factors associated with CPDI status; thus, this conversion might be considered an appropriate approach [60]. We also focused solely on Chinese immigrants with citizenship or permanent resident status, while Yang et al. (2024) extended their study population to all Chinese Canadians, including even visitors and international students [58]. Canadian citizens or permanent residents often have more privileges than temporary residents, including rights, freedoms, and connectivity n.d), which might contribute to variations in the results [61]. We chose a study population of citizens and permanent residents due to their more stable continuity of care, compared to temporary residents [61]. The pandemic could disrupt their access to healthcare, leading to psychological distress.

Our findings highlighted the influence of pandemic-related delayed medical care access on psychological distress among Chinese immigrants in Canada. These findings might be indicative of a broader range of issues regarding factors that may lead to delays in access or discourage seeking healthcare services, as well as potential impact on immigrants’ health. In fact, many factors such as language barriers, transportation barriers, long wait time, inadequate understanding of the health system, and the referral process might lead to delayed access or reluctance to seek care among immigrants [62,63]. However, a comprehensive understanding of their influences at each point of contact with the healthcare system and their impacts on psychological distress among immigrants is required.

Important policy implications can arise from our findings. The increased demand for mental health support during the pandemic highlights the importance of incorporating mental health services into emergency response and public health preparations [64]. Mental health services are often underused among immigrants due to a lack of public coverage and disrupted care during the pandemic [64]. To increase access to and utilization of psychological services, federal and provincial governments should provide funding to implement new tele-psychological programs and establish billing codes for these services [64,65]. These services should be implemented at the national level, ensuring that immigrants living across Canada have equal access to mental health systems. Although the pandemic has ended, the demands for mental health support may continue to rise. Sustainability for these publicly funded programs should be ensured. Additionally, allowing allied healthcare providers to bill under public health insurance plans when delivering mental health services can increase access and utilization among immigrants, as most residents in Canada contact their primary care providers for mental-health needs [64].

Furthermore, long-term public health restrictions might lead to layoffs, disrupted health services, prolonged wait times, or increased levels of uncertainty, contributing to psychological distress [66]. Immigrants were particularly prone to psychological distress during the pandemic [16,17,18,19,20]. Community health and social networks are important sources for immigrants to access information and receive support [66]. They should remain open to support immigrants, rather than temporarily closing during a pandemic [66]. Additionally, because of language barriers, public health guidelines or protocols should be made available in the languages of immigrants to ensure they understand and have access to services they need [66].

## 5. Strengths and Limitations

Our study provides important evidence showing how delayed access to treatment/health affected psychological distress in Chinese immigrants in Canada during the pandemic. Our study also confirms the strong negative impact of COVID-19 phobia and social loneliness on mental well-being in this population.

The literature shows that there are relationships among discrimination, sociodemographic backgrounds, delayed access to care, and loneliness [20,45,58], which might lead to a potential multicollinearity among covariates in the logistic regression model. Logistic LASSO regression could address this issue in our model-building.

The current study has some limitations. First, most participants were from Ontario, which might not represent the total Chinese population in Canada. Second, data collection through an online survey might pose selection bias, with individuals who do not have an internet connection or do not use electronic gadgets being under-represented in this study sample. However, an online survey was the appropriate approach for us to collect data during the pandemic. Third, this study may suffer from information bias (e.g., social desirability or recall bias) as the nature of self-report measures. The convenience sampling procedure also limits the representation of the target population. Due to using convenience sampling, the study’s findings might not be generalized to the total population of Chinese immigrants in Canada.

## 6. Conclusions

Fear of COVID-19, social loneliness, and delayed access to treatment/health assessment are risks for psychological distress among Chinese immigrants in Canada amidst the COVID-19 pandemic. A broader range of issues regarding factors that may result in delayed access or discourage seeking care, and their impact on psychological distress among immigrants, must be evaluated.

Public health agencies and policy makers need to handle these detrimental effects on the mental well-being of vulnerable immigrants. Efforts can include reliable health information from credible sources, mental health support and psychological programs, and virtual care tailored to immigrants that are available and accessible for these populations.

## Figures and Tables

**Table 1 healthcare-12-01639-t001:** Participants’ characteristics by CPDI levels.

Variables	N	CPDI		*p*-Value
		Normal (<28) (n = 419)	Mild-to-Severe (≥28) (n = 327)	
Age				0.7799
Under 35	51 (6.84)	26 (6.21)	25 (7.65)	
35–44	96 (12.87)	52 (12.41)	44 (13.46)	
45–54	265 (35.52)	146 (34.84)	119 (36.39)	
55–64	194 (26.01)	111 (26.49)	83 (25.38)	
65 or above	140 (18.77)	84 (20.05)	56 (17.13)	
Gender				0.0165
Men	251 (34.15)	156 (37.86)	95 (29.41)	
Women	484 (65.85)	256 (62.14)	228 (70.59)	
Birthplace				0.0074
Mainland China	710 (95.17)	391 (93.32)	319 (97.55)	
Other	36 (4.83)	28 (6.68)	8 (2.45)	
Mother language				0.7580
Mandarin	694 (93.53)	389 (93.29)	305 (93.85)	
Other	48 (6.47)	28 (6.71)	20 (6.15)	
Province				0.7181
Ontario	661 (89.08)	373 (89.45)	288 (88.62)	
Other	81 (10.92)	44 (10.55)	37 (11.38)	
Length in Canada				0.1976
Under 6 years	79 (10.59)	41 (9.79)	38 (11.62)	
6 to less than 15 years	238 (31.90)	125 (29.83)	113 (34.56)	
At least 15 years	429 (57.51)	253 (60.38)	176 (53.82)	
Marital status				0.2670
Married/Common law	643 (86.54)	366 (87.77)	277 (84.97)	
Other	100 (13.46)	51 (12.23)	49 (15.03)	
Living status				0.4310
With family	679 (91.14)	384 (91.87)	295 (90.21)	
Other	66 (8.86)	34 (8.13)	32 (9.79)	
Highest educational status				0.1327
Under university	165 (22.12)	84 (20.05)	81 (24.77)	
University	322 (43.16)	178 (42.48)	144 (44.04)	
Master/PhD	259 (34.72)	157 (37.47)	102 (31.19)	
Employment status				0.2767
Employed	312 (41.82)	183 (43.68)	129 (39.45)	
Self-employed	141 (18.90)	74 (17.66)	67 (20.49)	
Unemployed	105 (14.08)	52 (12.41)	53 (16.21)	
Other	188 (25.20)	110 (26.25)	78 (23.85)	
Work in healthcare				0.5312
Yes	67 (9.05)	40 (9.64)	27 (8.31)	
No	673 (90.95)	375 (90.36)	298 (91.69)	
Work requiring public contact				0.1215
Yes	132 (17.79)	66 (15.87)	66 (20.25)	
No	610 (82.21)	350 (84.13)	260 (79.75)	
Financial status				<0.0001
Dissatisfied	173 (23.22)	71 (16.99)	102 (31.19)	
Neutral	305 (40.94)	168 (40.19)	137 (41.90)	
Satisfied	267 (35.84)	179 (42.82)	88 (26.91)	
Current health status				0.0001
Poor	32 (4.30)	9 (2.15)	23 (7.06)	
Fair	299 (40.19)	154 (36.84)	145 (44.48)	
Good	413 (55.51)	255 (61.00)	158 (48.47)	
COVID-19 infectious history				1.0000 ^a^
Yes	4 (0.54)	2 (0.48)	2 (0.61)	
No	742 (99.46)	417 (99.52)	325 (99.39)	
Fear of COVID-19				<0.0001
Yes	537 (71.98)	254 (60.62)	283 (86.54)	
No	209 (28.02)	165 (39.38)	44 (13.46)	
Discriminate				0.0774
Perceived/experienced	286 (38.34)	149 (35.56)	137 (41.90)	
Other	460 (61.66)	270 (64.44)	190 (58.10)	
Delayed medical care access				<0.0001
Yes	600 (80.43)	314 (74.94)	286 (87.46)	
Non-response about delay	146 (19.57)	105 (25.06)	41 (12.54)	
Social Loneliness Score				
N	713	411	302	
Mean		16.21 ± 2.73	19.18 ± 2.98	<0.0001

Note. Distribution of categorical covariate variables between CPDI groups was compared using Chi-square tests (or^a^ Fisher’s exact test when more than 20% of cells with expected cell counts less than five). Mean scores of social loneliness in both groups were compared using the *t*-test. Abbreviations; CPDI = COVID-19 Peritraumatic Distress Index.

**Table 2 healthcare-12-01639-t002:** Risk factors associated with mild to severe CPDI level from the Univariate logistic regression model.

Variables	CPDI		*p*-Value
	OR	95% CI	
Age			
Under 35	1.442	0.757–2.748	0.4354
35–44	1.269	0.751–2.145	0.7636
45–54	1.223	0.807–1.853	0.8957
55–64	1.122	0.721–1.745	0.6249
65 or above	ref		
Gender			
Men	0.684	0.501–0.933	0.0167
Women	ref		
Birthplace			
Mainland China	2.855	1.284–6.352	0.0101
Other	ref		
Mother language			
Mandarin	1.097	0.606–1.985	0.7594
Other	ref		
Province			
Ontario	0.918	0.577–1.459	0.7171
Other	ref		
Length in Canada			
Under 6 years	1.332	0.823–2.156	0.5149
6 to less than 15 years	1.299	0.945–1.788	0.5072
At least 15 years	ref		
Marital status			
Married/Common law	0.788	0.517–1.201	0.2678
Other	ref		
Living status			0.4316
With family	0.816	0.492–1.354	
Other	ref		
Highest educational status			
Under university	1.484	1.001–2.201	0.1076
University	1.245	0.893–1.736	0.8846
Master/PhD	ref		
Employment status			
Employed	0.994	0.688–1.435	0.1700
Self-employed	1.277	0.823–1.982	0.5119
Unemployed	1.437	0.889–2.232	0.1825
Other	ref		
Work in healthcare			0.5316
Yes	0.849	0.509–1.416	
No	ref		
Work requiring public contact			0.1223
Yes	1.346	0.923–1.963	
No	ref		
Financial status			
Dissatisfied	2.922	1.966–4.342	<0.0001
Neutral	1.659	1.180–2.332	0.8440
Satisfied	ref		
Current health status			
Poor	4.123	1.860–9.135	0.0026
Fair	1.520	1.124–2.054	0.2149
Good	ref		
COVID-19 infectious history			
Yes	1.283	0.180–9.158	0.8037
No	ref		
Fear of COVID-19			
Yes	4.178	2.877–6.068	<0.0001
No	ref		
Discriminate			
Perceived/experienced	1.307	0.971–1.759	0.0777
No	ref		
Delayed medical care access			
Yes	2.333	1.572–3.462	<0.0001
Non-response about delay	ref		
Social Loneliness Score	1.447	1.354–1.545	<0.0001

Abbreviations: CPDI = COVID-19 Peritraumatic Distress Index; 95% CI = 95% Confidence interval; OR = Odds ratios.

**Table 3 healthcare-12-01639-t003:** Risk factors associated with mild-to-severe CPDI level from the multivariate logistic regression model.

Variables	Completed Cases Analysis			Imputed Cases Analysis		
	OR	95% CI	*p*-Value	OR	95% CI	*p*-Value
Age						
Under 35	1.946	0.858–4.411	0.3016	1.317	0.734–2.363	0.3551
35–44	1.534	0.802–2.934	0.7602	1.111	0.723–1.706	0.6307
45–54	1.510	0.897–2.540	0.7458	1.065	0.780–1.455	0.6899
55–64	1.343	0.771–2.339	0.7201	0.952	0.666–1.361	0.7867
65 or above	ref			ref		
Length in Canada						
Under 6 years	1.339	0.771–2.339	0.6733	1.108	0.759–1.616	0.5961
6 to less than 15 years	1.403	0.924–2.129	0.3770	1.120	0.844–1.486	0.4307
At least 15 years	ref			ref		
Highest educational status						
Under university	1.303	0.793–2.143	0.2299	1.170	0.880–1.556	0.2797
University	0.999	0.661–1.510	0.4690	0.943	0.743–1.196	0.6263
Master/PhD	ref			ref		
Financial status						
Dissatisfied	1.314	0.803–2.150	0.3736	1.133	0.852–1.507	0.3910
Neutral	1.168	0.776–1.758	0.9176	1.004	0.792–1.272	0.9757
Satisfied	ref			ref		
Fear of COVID-19						
Yes	2.617	1.694–4.041	<0.0001	1.604	1.293–1.989	<0.0001
No	ref			ref		
Delayed medical care access						
Yes	1.900	1.184–3.048	0.0078	1.362	1.078–1.720	0.0095
Non-response about delay	ref			ref		
Social Loneliness Score	1.384	1.291–1.483	<0.0001	1.408	1.314–1.508	<0.0001
Work requiring public contact						
Yes	1.245	0.782–1.982	0.3563	-	-	-
No	ref			-	-	-
Current health status						
Poor	-	-	-	1.529	0.815–2.869	0.1860
Fair	-	-	-	0.782	0.544–1.124	0.1838
Good	-	-	-	ref		

Abbreviations: CPDI = COVID-19 Peritraumatic Distress Index; 95% CI = 95% Confidence interval; OR = Odds ratios.

## Data Availability

The datasets used and/or analysed during this study are available from the corresponding author on reasonable request.

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
