# Peer review of "Delayed Access to Medical Care and Psychological Distress among Chinese Immigrants in Canada during the Pandemic"

_healthcare, 2024, doi:10.3390/healthcare12161639_

Round 1

Reviewer 1 Report

Comments and Suggestions for Authors

This paper seeks to address some of the challenges for migrant groups in Canada during the COVID-19 period. There is a potential for this paper, but it requires some major revisions particularly in relation to:

1. Research objectives and questions-These should be clearly stated in the early part of the paper. 

2. The literature should also center the scholarship on migrant populations to demonstrate some of the concerns in host countries like North America and the vulnerabilities that unfolded during the COVID-19 period. This will locate the relevance of the study and the focus on this specific population.

3. Methodology-It would be helpful to share with the reader who persons were recruited for the study. The data collection process and the randomization around this would be important aspects of the discussion to include.

4. Discussion and Limitations-There is little engagement of the findings in relation to the broader field/scholarship. This is important to help the reader situate the contributions that are made through this work and the limitations of the study. It would be also useful to discuss the theoretical, empirical and practical implications of the findings. 

Comments on the Quality of English Language

No major issues cited

Author Response

Dear Reviewer 1, 

Thank you very much for taking the time to review this manuscript. Please find the detailed responses below and the corresponding revisions/corrections highlighted in yellow in the re-submitted files

Comment 1: Research objectives and questions-These should be clearly stated in the early part of the paper

Response 1: Thank you for your suggestion. We have re-written it in the final paragraph of the paper (line.88-94)  

Comment 2: The literature should also center the scholarship on migrant populations to demonstrate some of the concerns in host countries like North America and the vulnerabilities that unfolded during the COVID-19 period. This will locate the relevance of the study and the focus on this specific population.

Response 2: Thank you for your suggestion. We have added one paragraph in the introduction section (line. 65-76)  

Comment 3: Methodology-It would be helpful to share with the reader who persons were recruited for the study. The data collection process and the randomization around this would be important aspects of the discussion to include.

Response 3: Thank you for your suggestion. Study participants, study samples, sampling method, and data collection process were mentioned in our original version. I have highlighted this paragraph in this revised version (line. 115-121). We have added one paragraph to clarify our recruiting process (line. 106-114)  

Comment 4: Discussion and Limitations-There is little engagement of the findings in relation to the broader field/scholarship. This is important to help the reader situate the contributions that are made through this work and the limitations of the study. It would be also useful to discuss the theoretical, empirical and practical implications of the findings.  

Response 4: Thank you for your suggestion. We have added one paragraph (400-409) about a broader range of issues regarding factors that may potentially cause delayed access to care among immigrants as well as a need for further studies to evaluate its impact on immigrants’ mental health  

Thank you for your time and consideration.  

Best regards,   

Dr. P. Peter Wang, MD, PhD, Professor (Epidemiology)

Division of Population Health and Applied Health Sciences

Faculty of Medicine, Memorial University of Newfoundland

300 Prince Philip Drive, St. John's, NL A1B 3V6

Reviewer 2 Report

Comments and Suggestions for Authors

This manuscript reports findings from an examination of delayed access to care or assessment on psychological distress among Chinese immigrants in Canada during the second wave of the COVID pandemic. Data were collected from a convenience sample of Chinese residing in Canada. Respondents were limited to permanent residents and Chinese Canadian citizens over the age of 16. The final usable sample was 746, from an original pool of 1288 respondents. Data were collected during January and February 2021. Objectives were to describe characteristics of Chinese immigrants by psychological distress levels and examine the association betwen delayed access to treatment or health assessments and levels of distress.

The measures used are appropriate to the questions being posed. For example, use of the De-Jong Loneliness Scale is appropriate. The statistical analysis (univariate logistic regression models, logistic LASSO regression, and multivariate regression) is appropriate.

Some minor editing for English grammar is needed:

Lines 54-56: The sentence that begins "In addition to various factors.... that resulted for.." should read "In addition to various factors...that resulted in deteriorating psychological status...

Line 94: It is better to use the term "convenience sample" rather than "convenient sample."

Lines 159-160: The sentence that begins "Participants who respondents...those who did not respond any changes..." should read "...those who did not report any changes..."

Lines 163-165: The sentence that begins "All missing data...a valid approach to handle with missing..." should read "...valid approach to handle missing.."

Line 186: It is better to use the term "t-test" rather than "ttest."

Lines 189-190: The sentence that begins "Subsequently, for sensitivity analysis...logistic models for completed dataset.." should read "...logistic models for the completed dataset..."

Line 191: The sentence that begins "We used logistic LASSO regression to select most significant variables..." should read "...to select the most significant variables..."

Lines 255-256: The sentence that begins "Table 3 presents risk factors...from multivariate regression model" should read "...from the multivariate regression model."

Lines 286-287: The sentence that begins "Jonathan..can correct biases that cause by by the CCA" should read "...can correct biases caused by the CCA."

Lines 333-337: The sentence that begins "Mehrunnisa et al...found that..." should read "...during the pandemic results in poorer mental health...but the specific impact..." The next sentence that begins "Our findings help demonstrate extend..." is unclear and needs to be rewritten.

Line 340-341: The sentence tha begins "However, Patrick...were barriers of implementing" should read "However,...should read either "barriers in implementing" or "barriers to implementing..."

Lines 359-360: The sentence that begins "Our study provided...access to treatment/health assessment on psychological..." should read "Our study provided...assessment affected..."

Lines 373-374: The sentence that begins, "Third, this study can suffer information bias...as the nature..." should read "...due to the nature..." The following sentence that begins "Convenient sampling procedure also limits..." should read "The convenience sampling procedure..."

Lines 375-76: The sentence that begins "Therefore, the study's findings should be interpreted cautiously...generalize the results for the total Chinese immigrant in Canada" should be rewritten. Due to the use of a convenience sample the findings can not be generalized to the total population of Chinese immigrants in Canada. This should be stated more directly.

Other Comments:

The subheading "4.0. Discussion" is missing. It should be placed somewhere around line 297.

Authors state that the participants were recruited using a convenience sampling procedure. This is insufficient information. We need more information about how they were recruited. Advertisements in Chinese language media? Flyers in health clinics? Recruiting through Facebook or other social media?

The Chinese immigrants are described as a vulnerable population (line 73). Why are they vulnerable? One or two sentences should be added to explain this for the reader.

The authors note that they failed to find an association between perceived or experienced discrimination and distress, unlike other studies they noted. Why not? The authors should address this with a couple of sentences (Lines 352-355). Was this population more privileged/better integrated in some way than populations studied by Lixia et al. (2022) for example?

Comments on the Quality of English Language

Minor editing needed. This is noted in my comments.

Author Response

Dear Reviewer 2, 

Thank you very much for taking the time to review this manuscript. Please find the detailed responses below and the corresponding revisions/corrections highlighted in yellow in the re-submitted files.  

Comment 1: Lines 54-56: The sentence that begins "In addition to various factors.... that resulted for.." should read "In addition to various factors...that resulted in deteriorating psychological status...

Response 1: Thank you so much. We have changed (line 57-58)  

Comment 2: Line 94: It is better to use the term "convenience sample" rather than "convenient sample."

Response 2: Thank you so much. We have changed (line. 106)  

Comment 3: Lines 159-160: The sentence that begins "Participants who respondents...those who did not respond any changes..." should read "...those who did not report any changes..."

Response 3: Thank you so much. We have changed (line. 179-180)  

Comment 4: Lines 163-165: The sentence that begins "All missing data...a valid approach to handle with missing..." should read "...valid approach to handle missing.."

Response 4: Thank you so much. We have changed (line. 185)  

Comment 5: Line 186: It is better to use the term "t-test" rather than "ttest."

Response 5: Thank you so much. We have changed (line. 206)  

Comment 6: Lines 189-190: The sentence that begins "Subsequently, for sensitivity analysis...logistic models for completed dataset.." should read "...logistic models for the completed dataset..."

Response 6: Thank you so much. We have changed (line. 209-210)  

Comment 7: Line 191: The sentence that begins "We used logistic LASSO regression to select most significant variables..." should read "...to select the most significant variables..."

Response 7: Thank you so much. We have changed (line. 211)  

Comment 8: Lines 255-256: The sentence that begins "Table 3 presents risk factors...from multivariate regression model" should read "...from the multivariate regression model."

Response 8: Thank you so much. We have changed (line. 278-279)  

Comment 9: Lines 286-287: The sentence that begins "Jonathan..can correct biases that cause by by the CCA" should read "...can correct biases caused by the CCA."

Response 9: Thank you so much. We have changed (line. 310)  

Comment 10: Lines 333-337: The sentence that begins "Mehrunnisa et al...found that..." should read "...during the pandemic results in poorer mental health...but the specific impact..." The next sentence that begins "Our findings help demonstrate extend..." is unclear and needs to be rewritten.

Response 10: Thank you so much. We have changed (line. 363-366)  

Comment 11: Line 340-341: The sentence tha begins "However, Patrick...were barriers of implementing" should read "However,...should read either "barriers in implementing" or "barriers to implementing..."

Response 11: Thank you so much. We have changed (line. 372)  

Comment 12: Lines 359-360: The sentence that begins "Our study provided...access to treatment/health assessment on psychological..." should read "Our study provided...assessment affected..."

Response 12: Thank you so much. We have changed (line 411-412)  

Comment 13: Lines 373-374: The sentence that begins, "Third, this study can suffer information bias...as the nature..." should read "...due to the nature..." The following sentence that begins "Convenient sampling procedure also limits..." should read "The convenience sampling procedure..."

Response 13: Thank you so much. We have changed (line 426-429)  

Comment 14: Lines 375-76: The sentence that begins "Therefore, the study's findings should be interpreted cautiously...generalize the results for the total Chinese immigrant in Canada" should be rewritten. Due to the use of a convenience sample the findings can not be generalized to the total population of Chinese immigrants in Canada. This should be stated more directly.

Response 14: Thank you so much. We have changed (line. 426-429)  

Comment 15: The subheading "4.0. Discussion" is missing. It should be placed somewhere around line 297.

Response 15: We are so sorry for this inconvenience. The “4. Discussion” was hidden under the footnote below table 3. We have retrieved it.  

Comment 16: Authors state that the participants were recruited using a convenience sampling procedure. This is insufficient information. We need more information about how they were recruited. Advertisements in Chinese language media? Flyers in health clinics? Recruiting through Facebook or other social media?

Response 16: Thank you for your suggest. We have added information requested in line 106 - 114  

Comment 17: The Chinese immigrants are described as a vulnerable population (line 73). Why are they vulnerable? One or two sentences should be added to explain this for the reader.

Response 17: Thank you for your suggest. I have added it into a new paragraph from line 66-76  

Comment 18: The authors note that they failed to find an association between perceived or experienced discrimination and distress, unlike other studies they noted. Why not? The authors should address this with a couple of sentences (Lines 352-355). Was this population more privileged/better integrated in some way than populations studied by Lixia et al. (2022) for example?

Response 18: Thank you for your great question. I have added a paragraph in the discussion to answer your question (line 390-399).
“Variations in the findings might result from different analytic decisions, the number of covariates, and the chosen study sample (Silberzahn et al., 2018). For example, we focused solely on Chinese immigrants with citizenship or permanent resident status while Yang et al. (2024) extended their study population to all Chinese Canadians, including even visitors and international students. Canadian citizens or permanent residents often have more privileges than temporary residents, including rights, freedoms, and connectivity (Canada’s Permanent Residence Card: A Newcomer’s Guide, n.d), which might contribute to variations in the results. We chose a study population of citizens or permanent residents due to their more stable continuity of care, compared to temporary residents. The pandemic could disrupt their access to healthcare, leading to psychological distress”  

Thank you for your time and consideration.   

Best regards,   

Dr. P. Peter Wang, MD, PhD, Professor (Epidemiology)

Division of Population Health and Applied Health Sciences

Faculty of Medicine, Memorial University of Newfoundland

300 Prince Philip Drive, St. John's, NL A1B 3V6

Reviewer 3 Report

Comments and Suggestions for Authors

Dear Editor. Thank you for the opportunity to read this paper.

It is a very interesting, well constructed and well thought out paper. I particularly liked the methodology.

There are small changes suggested that I think can improve the article.

1) In methodology, summarize the content of section 2.2. Missing data. Place the content (summarized) at the end of section 2.1. Study design, setting, and participants.

2) It is recommended that the presentation of the tables in the text be done with a mention of the table according to its numbering and not with "the table shown in...", or "table number xxx contains".

I have missed some figure illustrating the discussion section.

3) The discussion section is a discussion of the results found.

It is necessary to include references to the results and discuss them. It is proposed to include subheadings that make explicit reference to the research objectives.

4) Citations and references.

First, it is necessary to follow the journal's instructions for citations and references.

Secondly, check thoroughly that there are no citations without references, nor references without citations. I have not stopped to see if there are more, but almost at the beginning of the article there is a citation that has no reference (or at least I have not been able to find it): Gibney et al., 2022.

Author Response

Dear reviewer 3, 

Thank you very much for taking the time to review this manuscript. Please find the detailed responses below and the corresponding revisions/corrections highlighted yellow in the re-submitted files.   

Comment 1: In methodology, summarize the content of section 2.2. Missing data. Place the content (summarized) at the end of section 2.1. Study design, setting, and participants

Response 1: Thank you for your comment, and it is helpful. However, to maintain the paper’s length and improve convenience for readers, we have provided a summary of our data, including missing data in Supplemental material - S1  

Comment 2: It is recommended that the presentation of the tables in the text be done with a mention of the table according to its numbering and not with "the table shown in...", or "table number xxx contains".

Response 2: We have changed (line. 222-223), (line. 255-256), (line. 266-267), (line. 278-279)  

Comment 3: I have missed some figure illustrating the discussion section.The discussion section is a discussion of the results found. It is necessary to include references to the results and discuss them. It is proposed to include subheadings that make explicit reference to the research objectives.

Response 3: Thank you for your suggestion and it’s very helpful. We have changed (line. 324, 347, 361, 380)  

Comment 4: Citations and references.First, it is necessary to follow the journal's instructions for citations and references.Secondly, check thoroughly that there are no citations without references, nor references without citations. I have not stopped to see if there are more, but almost at the beginning of the article there is a citation that has no reference (or at least I have not been able to find it): Gibney et al., 2022.

Response 4: Thank you for your queries. Please accept the apologizes for this mistake. Our first author’s reference manager set up authors’ first name in text-citations and their last name in the reference list. Therefore, you noticed that no citations in-text were in the reference list. We have corrected this mistake. We have indicated locations of original in-text citations in the reference list attached in the cover letter.

Thank you for your time and consideration. 

Best regards, 

Dr. P. Peter Wang, MD, PhD, Professor (Epidemiology)

Division of Population Health and Applied Health Sciences

Faculty of Medicine, Memorial University of Newfoundland

300 Prince Philip Drive, St. John's, NL A1B 3V6

Reviewer 4 Report

Comments and Suggestions for Authors

The WHO has already published findings on the impact of COVID-19 on mental health. There is currently a lack of research on the impact of the interventions that have been implemented. However, more attention and efforts are needed to ensure access to mental health.

The title indicates the content of the study. But it is too long. Keep it to 15 words.

The abstract allows quick identification of the basic content, describes the objective, methodology, results, but the conclusions are missing. Eliminate acronyms (CPDI).

Several similar articles published by some of the authors have been found, some are presented here:

˗          Yang, L., Yu, L., Kandasamy, K., Wang, Y., Shi, F., Zhang, W., & Wang, P. P. (2022, November). Non-pathological psychological distress among Mainland Chinese in Canada and its sociodemographic risk factors amidst the pandemic. In HealthCare (Vol. 10, No. 11, p. 2326). MDPI.

˗          Vo, A. T., Yang, L., Urquhart, R., Yi, Y., & Wang, P. P. (2024). The Impact of Delayed Access to Care on Psychological Distress Among Chinese Immigrants in Canada During the Second Wave of the Pandemic.

˗          Yang, L., Kandasamy, K., Na, L., Zhang, W., & Wang, P. (2024). Perceived and experienced anti-Chinese discrimination and its associated psychological impacts among Chinese Canadians during the Wave 2 of the COVID-19 pandemic. Psychology, Health & Medicine29(1), 108-125.

It has the same objective, methodology and results as the one presented, most of the authors are the same. It appears to be a large study that was fragmented into several articles for publication.

The review shows the what and why of the research during the pandemic, although it is not clear how useful the study is now, three years later. It is also unclear whether the references are correct, as only eight of the 53 references in the list match those in the text. References in the text and list should be written the same way and match (e.g. in the text Lixia, Kesaan, Ling et al, 2022 and in the list Yang, L., Yu, L., Kandasamy, K., Wang, Y., Shi, F., Zhang, W., & Wang, P. P. (2022).

Several articles by the same authors in which they publish similar content. Several self citations under different names.

The structure of the article and the arguments are logical and coherent, although the steps in the data collection procedure are not clearly described.

When an acronym is introduced for the first time, it is necessary to describe what it means, e.g: Intensive Care Unit (ICU).

The methodology seems appropriate and in line with the theoretical justification. Explain the convenience sampling procedure and how the questionnaires were sent out.

Explain the ethical considerations that have been taken into account.

The data are analysed in relation to the aim of the study. Lines 292, 293 and 294 are missing from the results, so the following sentence does not provide any information. The results discussed with other similar studies are not clear as the references cannot be found. It should suggest future research on the problem, especially considering that the study is three years old. The conclusions are brief and do not add anything new to other published studies by the same authors.

References are up to date. The references in the text are not the same as those in the list, and it is not known whether they refer to the same authors. If there are several references in the text, check that they are listed in alphabetical order. For example: (Nicole et al, 2021; Mehrunnisa et al, 2024) should be (Mehrunnisa et al, 2024; Nicole et al, 2021).

Author Response

Dear Reviewer 4, 

Thank you very much for taking the time to review this manuscript. Please find the detailed responses below and the corresponding revisions/corrections highlighted in yellow in the re-submitted files.   

Comment 1: The WHO has already published findings on the impact of COVID-19 on mental health. There is currently a lack of research on the impact of the interventions that have been implemented. However, more attention and efforts are needed to ensure access to mental health.

Response 1: I agree with your comment. Particularly, we appreciate your insights on efforts needed to ensure access to mental health for immigrant, which are very helpful. Although impacts of COVID-19 on mental health might be well-known, some potential factors can be continued to be consider into the broader issues. Our study emphasized not only COVID-19-related factors impacting mental health, but also aimed to highlight delayed access to care on psychological distress among immigrants. Current study focused on COVID-19-related delated access to care, so we will continue to expand to other factors in future studies.  

Comment 2: The title indicates the content of the study. But it is too long. Keep it to 15 words.

Response 2: Thank you so much. We have changed  

Comment 3: The abstract allows quick identification of the basic content, describes the objective, methodology, results, but the conclusions are missing. Eliminate acronyms (CPDI).

Response 3: Thank you so much. In abstract, we have added one sentence for conclusion, and explained acronyms what CPDI is.  

Comment 4: Several similar articles published by some of the authors have been found, some are presented here:

˗          Yang, L., Yu, L., Kandasamy, K., Wang, Y., Shi, F., Zhang, W., & Wang, P. P. (2022, November). Non-pathological psychological distress among Mainland Chinese in Canada and its sociodemographic risk factors amidst the pandemic. In HealthCare (Vol. 10, No. 11, p. 2326). MDPI.

˗          Vo, A. T., Yang, L., Urquhart, R., Yi, Y., & Wang, P. P. (2024). The Impact of Delayed Access to Care on Psychological Distress Among Chinese Immigrants in Canada During the Second Wave of the Pandemic

˗          Yang, L., Kandasamy, K., Na, L., Zhang, W., & Wang, P. (2024). Perceived and experienced anti-Chinese discrimination and its associated psychological impacts among Chinese Canadians during the Wave 2 of the COVID-19 pandemic. Psychology, Health & Medicine29(1), 108-125. 

It has the same objective, methodology and results as the one presented, most of the authors are the same. It appears to be a large study that was fragmented into several articles for publication.  

Response4: 

We want to verify that “Vo, A. T., Yang, L., Urquhart, R., Yi, Y., & Wang, P. P. (2024). The Impact of Delayed Access to Care on Psychological Distress Among Chinese Immigrants in Canada During the Second Wave of the Pandemic” is current study under your review. It was pre-printed but not peer reviewed yet.

Two articles

Yang, L., Yu, L., Kandasamy, K., Wang, Y., Shi, F., Zhang, W., & Wang, P. P. (2022, November). Non-pathological psychological distress among Mainland Chinese in Canada and its sociodemographic risk factors amidst the pandemic. In HealthCare (Vol. 10, No. 11, p. 2326). MDPI.

The study used the data collected in the first wave of COVID-19 while our data was collected in the second wave of COVID-19

Yang, L., Kandasamy, K., Na, L., Zhang, W., & Wang, P. (2024). Perceived and experienced anti-Chinese discrimination and its associated psychological impacts among Chinese Canadians during the Wave 2 of the COVID-19 pandemic. Psychology, Health & Medicine, 29(1), 108-125

Although we used the same data, the study objectives, study population, and study method were different. Yang et al. (2024) focused on identifying  perceived/experienced discrimination and its impact on psychological distress among Chinese Canadians. In contrast, we focused on delayed access to care and its impact on psychological distress among Chinese Canadians emphasizing solely on citizens and permanent residents. We chose this population because we assumed that they have more stable continuity of care, compared to temporary residents (e.g., visitors or international students). The pandemic could disrupt their access to healthcare, leading to psychological distress.

Comment 5: The review shows the what and why of the research during the pandemic, although it is not clear how useful the study is now, three years later. It is also unclear whether the references are correct, as only eight of the 53 references in the list match those in the text. References in the text and list should be written the same way and match (e.g. in the text Lixia, Kesaan, Ling et al, 2022 and in the list Yang, L., Yu, L., Kandasamy, K., Wang, Y., Shi, F., Zhang, W., & Wang, P. P. (2022).

Several articles by the same authors in which they publish similar content. Several self citations under different names.

Response 5: Thank you for your comment. As I mentioned, although impacts of COVID-19 on mental health might be well-known, some potential factors can be continued to be consider into the broader issues. Our study emphasized not only COVID-19-related factors impacting mental health, but also aimed to highlight delayed access to care on psychological distress among immigrants. Current study focused on COVID-19-related delated access to care, so we will continue to expand to other factors in future studies. We have added one paragraph to discuss about this in line 400-409.

In terms of references, please accept the apologizes for this mistake. Our first author’s reference manager set up authors’ first name in text-citations and their last name in the reference list. Therefore, you noticed that no citations in-text were in the reference list. We have corrected this mistake. We have indicated locations of original in-text citations in the reference list attached in this cover letter

Comment 6: The structure of the article and the arguments are logical and coherent, although the steps in the data collection procedure are not clearly described. 

Response 6: Thank you so much. We changed (line. 106-114)

Comment 7: When an acronym is introduced for the first time, it is necessary to describe what it means, e.g: Intensive Care Unit (ICU).

Response 7: Thank you so much. We changed (line. 40)

Comment 8: The methodology seems appropriate and in line with the theoretical justification. Explain the convenience sampling procedure and how the questionnaires were sent out. Explain the ethical considerations that have been taken into account.

Response 8: Thank you so much. We explained in detail about sampling, recruiting, and collecting data as well as ethical considerations in line 106-122 of the revised version 

Comment 9: The data are analysed in relation to the aim of the study. Lines 292, 293 and 294 are missing from the results, so the following sentence does not provide any information. The results discussed with other similar studies are not clear as the references cannot be found. It should suggest future research on the problem, especially considering that the study is three years old. The conclusions are brief and do not add anything new to other published studies by the same authors.

Response 9: We are so sorry for this inconvenience. Table 3 is presented from line 292, 293, 294 in the original version, followed by the “4. Discussion” that was hidden under the footnote below the table 3. We retrieved it. We also added one paragraph to suggest required future studies about on factors influencing delayed access to care and their impact on psychological distress among immigrants (line. 400-409)

Comment 10: References are up to date. The references in the text are not the same as those in the list, and it is not known whether they refer to the same authors. If there are several references in the text, check that they are listed in alphabetical order. For example: (Nicole et al, 2021; Mehrunnisa et al, 2024) should be (Mehrunnisa et al, 2024; Nicole et al, 2021).

Response 10: Thank you so much. We have corrected and updated the references. Some references in the list that have highlighted in yellow are those added in the revised version.

Thank you for your time and consideration. 

Best regards, 

Dr. P. Peter Wang, MD, PhD, Professor (Epidemiology)

Division of Population Health and Applied Health Sciences

Faculty of Medicine, Memorial University of Newfoundland

300 Prince Philip Drive, St. John's, NL A1B 3V6

Round 2

Reviewer 1 Report

Comments and Suggestions for Authors

This to a great improvement to the first submission and with clear articulation of the objectives, methods and findings. The paper therefore is generally publishable with the need for some minor improvements. It is recommended the authors strengthen the introduction to show the importance of looking at migrant groups during the pandemic and to also speak to the implications of the findings for policy and practice. In other words, how can these issues be address through policy implementation and institutional interventions? Some discussion on this will widen the relevance of this article for the readership.  

Comments on the Quality of English Language

NA

Author Response

Dear Reviewer, 

We would like to express our sincere gratitude for your valuable time in reviewing our manuscript and providing constructive feedback. We are pleased to hear that you find this version a great improvement, with clear articulation of the objectives, methods, and findings.

We hope that the manuscript carefully revised may meet your high standards. Below we provide the point-by-point responses. All modifications in the manuscript have been highlighted in blue.

Comment: This to a great improvement to the first submission and with clear articulation of the objectives, methods and findings. The paper therefore is generally publishable with the need for some minor improvements. It is recommended the authors strengthen the introduction to show the importance of looking at migrant groups during the pandemic and to also speak to the implications of the findings for policy and practice. In other words, how can these issues be address through policy implementation and institutional interventions? Some discussion on this will widen the relevance of this article for the readership.  

Response:

Per your suggestions, we have made the following revisions:

Introduction Section: We have added content to the introduction that briefly explains the importance of focusing on immigrant populations (see lines 89-90 and 97-101)

Discussion of Implications: Two new paragraphs have been added to discuss the implications of our findings, particularly in relation to policy implementation and institutional interventions (see lines 408-434).

Additional References: To strengthen our arguments, we have included five additional references, which are highlighted in blue in the revised manuscript.

Thank you for your time and consideration. 

Best regards, 

Peizhong Peter Wang, MD, PhD, Professor of Epidemiology 

Reviewer 4 Report

Comments and Suggestions for Authors

The manuscript is much improved. Although I would ask that the self-citations in the text be removed. 

Author Response

Dear Reviewer 

Thank you very much for taking the time to review this manuscript. Please find the detailed responses below and the corresponding revisions/corrections highlighted green in the re-submitted files.   

Comment: The manuscript is much improved. Although I would ask that the self-citations in the text be removed.

Response: 

We appreciate your comments in the first round, which made our manuscript much improved.

We agree with you that our manuscript cited four articles where the same group of researchers contributed.

Please let us explain in details which citations can be removed from the manuscript.

- In the study by Lee et at. (2022), the “COVID-19 Peritraumatic Distress Index (CPDI) Scale” was used to measure the psychological distress level in the Chinese Canadian population. We used the same scale and, consequently, cited the article. We also cited another study by Zhong et al., 2021, which utilized this scale in China. We agree to remove the reference to Lee et al. (2022) from both the text (line. 324-326) and the reference list (line. 1073-1074).

Additionally, we have removed references to Yang et al. (2022) and Yang et al. (2024) from the text in the introduction section (line 101-102). However, we will retain these citations in discussion section.

Please let us explain why citations for Yang et al. (2022), Yang et al. (2024) and Yu et al. (2022) need to be retained in the manuscript.

In the studies by Yang et al. (2022), Yang et al. (2024), and Yu et al. (2024), their findings might allow us to draw comparisons with ours regarding what is consistent and contrasts. In the first revision of the manuscript, we provided a paragraph explaining why our findings contrast with those of Yang et al. (2022) and Yang et al. (2024) (line 806-816). Moreover, these articles revealed relationships among factors, including discrimination, loneliness, and sociodemographic factors. These findings highlighted the importance of our method using LASSO regression model, which addresses potential multicollinearity in logistic regression models.

Thank you for your time and consideration.

Dr. P. Peter Wang, MD, PhD, Professor (Epidemiology)

Division of Population Health and Applied Health Sciences

Faculty of Medicine, Memorial University of Newfoundland

300 Prince Philip Drive, St. John's, NL A1B 3V